# Controlling energy levels and Fermi level en route to fully tailored energetics in organic semiconductors

Ross Warren [1*], Alberto Privitera[1], Pascal Kaienburg [1], Andreas E. Lauritzen[1], Oliver Thimm[2], Jenny Nelson [3] & Moritz K. Riede [1*]

Simultaneous control over both the energy levels and Fermi level, a key breakthrough for inorganic electronics, has yet to be shown for organic semiconductors. Here, energy level tuning and molecular doping are combined to demonstrate controlled shifts in ionisation potential and Fermi level of an organic thin film. This is achieved by p-doping a blend of two host molecules, zinc phthalocyanine and its eight-times fluorinated derivative, with tunable energy levels based on mixing ratio. The doping efficiency is found to depend on host mixing ratio, which is explained using a statistical model that includes both shifts of the host's ionisation potentials and, importantly, the electron affinity of the dopant. Therefore, the energy level tuning effect has a crucial impact on the molecular doping process. The practice of comparing host and dopant energy levels must consider the long-range electrostatic shifts to consistently explain the doping mechanism in organic semiconductors.

[1] Clarendon Laboratory, Department of Physics, University of Oxford, Parks Road, Oxford OX1 3PU, UK. [2] IEK5-Photovoltaics, Forschungszentrum Jülich, 52425 Jülich, Germany. [3] Department of Physics, Imperial College London, Exhibition Road, London SW7 2AZ, UK. *email: physics@rosswarren.net; moritz.riede@physics.ox.ac.uk

The dominance of digital electronics based on inorganic semiconductors was achieved by control over the semiconductor's energy levels in combination with independently tuning the Fermi level energy. In the field of organic electronics, such simultaneous control has yet to be demonstrated. Here, we take a step towards this goal by combining band structure engineering effects[1]—that is the tuning of a thin film's energy levels—with Fermi level control via molecular doping. Molecular doping is the introduction of impurities altering the electronic properties of a semiconductor over many orders of magnitude. For organic electronic devices, such as solar cells[2], light-emitting diodes (OLEDs)[3,4] and field-effect transistors[5,6], control of molecular doping has proven to be critical for energy level alignment at interfaces, and minimising Ohmic loss across devices[3,5,7]. Although commercially successful in OLED displays, the fundamental principles governing efficient doping in organic semiconductors remain a topic of debate.

The prevailing practice for selecting host and dopant combinations is based on securing a favourable offset between the host and dopant's energy levels, typically measured independently from one another. For p-doping, this would involve introducing a dopant with an electron affinity (EA) larger than the ionisation potential (IP) of the host semiconductor. However, as reported for the well-studied case of zinc-phthalocyanine (ZnPc) and molecular p-dopant $F_6$-TCNNQ[8–11], this understanding of the doping mechanism is incomplete. Judging from the energy levels of the isolated molecules[1,12,13], the doping process is expected to be more efficient than is experimentally observed[8].

Tuning the energetic offset between host and dopant can be achieved by molecular modification, through means such as halogenation[14,15]. However, this method is limited to discrete changes in energy. Recently the gradual tuning of the effective IP of an organic thin film of ZnPc and its 8 times fluorinated derivative $F_8$ZnPc was achieved by the mixing of molecules with different IPs[1]. The tuning was enabled by a relatively long-ranged interaction, extending over a few neighbouring molecules along the $\pi$-$\pi$-stacking direction, mediated via the molecules' opposing quadrupole moments[16]. As with molecular doping, energy level tuning was found to critically influence device performance, demonstrated by altering the open-circuit voltage of organic solar cells[1]. Employing both doping and energy level tuning effects simultaneously, would add a powerful degree of freedom to the engineering of functional optoelectronic devices.

In this study, we combine energy level tuning and molecular doping to achieve simultaneous control of the ionization potential and Fermi level in an organic ternary blend film. We identify the quadrupolar electrostatic interaction between the host and dopant molecules to be crucial for the dopant's effective energy level in the mixed film. Specifically, we demonstrate that this energy level

tuning effect can explain the low doping efficiencies observed in doped organic semiconductors. The ternary blend comprises co-evaporated ZnPc:$F_8$ZnPc as the host mixture, with p-dopant $F_6$-TCNNQ. We perform photothermal deflection spectroscopy (PDS) and electron paramagnetic resonance (EPR) on the ternary blend system, which establishes a clear trend of increasing doping efficiency as the ZnPc content of the blend increases. A model based on Fermi-Dirac statistics successfully reproduces our experimental trends, but only by taking into account the effects of the quadrupole interactions on the dopant's EA. This leads us to question the ubiquitous practice of selecting hosts and dopants based on favourable energy level offset, as measured in their pristine forms. Instead, we propose that the electrostatic shifts in energy due to charge-quadrupolar interactions must be considered. These results show that it is possible to simultaneously control the host semiconductor's IP and the thin film's Fermi energy, levelling the playing field against inorganic semiconductors.

## Results

**Photothermal deflection spectroscopy**. Previous experiments have shown that ground-state integer-charge transfer occurs for ZnPc:$F_6$-TCNNQ[8,12], leading to at least partial polaron pair formation on the host H → H$^+$ and dopant D → D$^-$. As the absorption coefficients of these sub-gap states are typically very low[17], we perform photothermal deflection spectroscopy (PDS) measurements at room temperature. Figure 1a shows the absorption coefficient measured by PDS against probe beam energy for blends of ZnPc:$F_8$ZnPc p-doped with 0.05 molar ratio (MR) of $F_6$-TCNNQ, over a range of host composition ratios (see Supplementary Fig. 1 for chemical structures). For all cases, the optical absorption coefficient shows a region of low absorption below the $\pi \to \pi^*$ transitions of the phthalocyanine's Q-band, with several features apparent in the sub-gap region. Firstly, there are two peaks at 1.06 eV (1170 nm) and 1.24 eV (1000 nm), as marked on Fig. 1a, which are attributed to the $F_6$-TCNNQ$^-$ anions[8,18]. Although the intensities of the anion peaks are largest for transitions in ZnPc:$F_6$-TCNNQ, there is evidence of charge transfer for all samples, even for $F_8$ZnPc:$F_6$-TCNNQ. The integer charge transfer occurs from the IP of the host to the EA of the dopant. For the case of pure ZnPc as the host, this energy offset is thought to be favourable and the doping effect is expected. In contrast, the IP of $F_8$ZnPc exceeds the EA of the dopant. It is therefore surprising that a peak at 1.06 eV of the ionised dopant is present at all with the host of pure $F_8$ZnPc.

To quantify the dopant absorption features, the peaks in the sub-gap region are fitted with Gaussian functions. The peak height of the fits are normalised to the $\pi \to \pi^*$ transition at

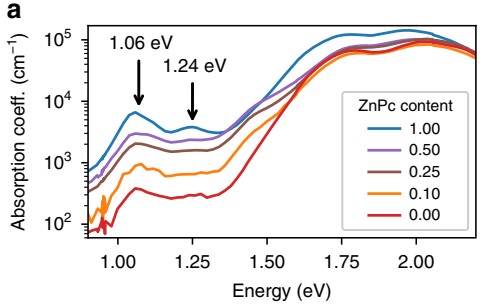
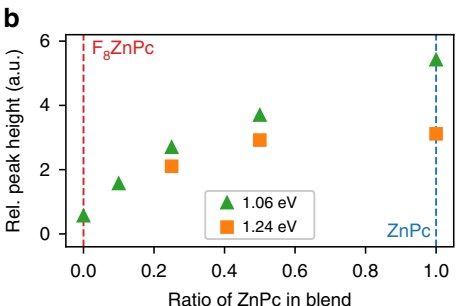

**Fig. 1** Absorption spectra of p-doped films. **a** Absorption spectra for different weight ratios of ZnPc:$F_8$ZnPc p-doped at 0.05 molar ratio (MR) with $F_6$-TCNNQ. The unmixed host of ZnPc with the dopant is marked as 1.00 and the unmixed case of $F_8$ZnPc as host is marked as 0.00. **b** Relative peak height of the absorption features at 1.06 eV (1170 nm) and 1.24 eV (1000 nm), against the weight ratio of ZnPc in the host blend. These features are attributed to the ionised dopant. The peak heights are scaled relative to the intensity of the Q-band absorption peak at 1.77 eV (701 nm).

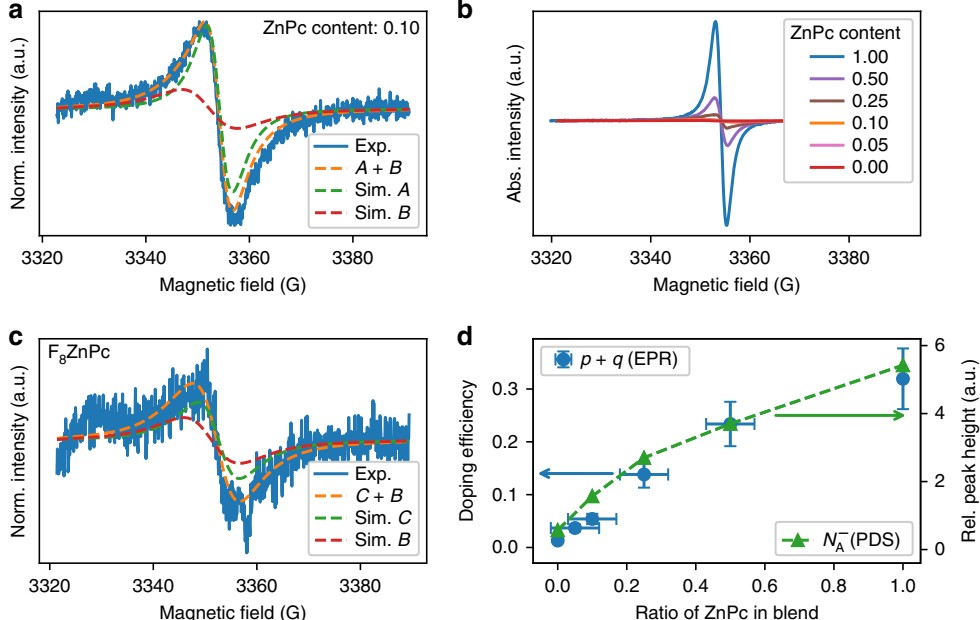

**Fig. 2** EPR spectra and analysis of p-doped films. Continuous wave EPR measurements, in the dark and at room temperature, on blends p-doped with 0.05 MR of $F_6$-TCNNQ. **a** EPR spectra of ZnPc:$F_8$ZnPc at weight ratio of 0.10:0.90. The dashed orange lines represent the spectral simulations of the EPR spectrum obtained as the sum of two contributions from the ZnPc positive polaron (species A, green dashed line) and the $F_6$-TCNNQ anion (species B, red dashed line). **b** EPR spectra of all the blend ratios. **c** EPR spectra of $F_8$ZnPc. The positive polaron is marked as species C as it has a larger $g$-value. **d** Doping efficiency $(p + q)/N_A$ (blue circles, left axis), as calculated from the EPR signal, plotted against weight ratio of ZnPc in the blend. The relative peak intensity $N_A^-$ from the PDS measurements are also plotted (right axis, green triangles). The error bars are propagated from the uncertainty in the spectral simulations and material deposition rates.

1.77 eV (701 nm) and are plotted in Fig. 1b against the ratio of ZnPc in the host blend. The absorption coefficient shows a clear monotonic increase as the content of ZnPc in the blend increases. As the doping concentration is fixed, the trend in the absorption coefficient is proportional to the fraction of ionised dopants $N_A^-/N_A$. This trend indicates that charge transfer from host to dopant is dependent on the blend composition. More precisely, a greater fraction of dopants ionise when the ZnPc content in the blend increases.

There are two further interesting features in the PDS spectra. Firstly, the Q-band transition of the $F_8$ZnPc is blue-shifted by around 20 meV relative to the ZnPc. This shift has also been reported by Brendel et al. and can be attributed to changes in the transition dipole alignment and intermolecular coupling upon fluorination[19]. Furthermore, this blue-shift is pertinent for all mixed cases, suggesting that the blend favours a structure with greater similarity to that of $F_8$ZnPc over that of the ZnPc. This observation is supported by $d$-spacing measurements from grazing-incidence wide-angle scattering (GIWAXS) images (see Supplementary Note 1, Supplementary Table 1 and Supplementary Fig. 2). Secondly, there is a broad absorption feature present around 1.5 eV in all cases of the ZnPc:$F_8$ZnPc blend. Similar absorption features have been indicative of ground-state charge transfer complexes[20], and could represent the ZnPc$^+$ cation, which has been reported to have an absorption feature at 1.48 eV (840 nm)[21]. Deeper analysis of this feature is beyond the scope of this investigation.

**Electron paramagnetic resonance spectroscopy**. The ground-state integer-charge transfer between the host and dopant, as observed in the PDS, generates unpaired electrons. The unpaired spins can be studied using electron paramagnetic resonance (EPR) spectroscopy. Figure 2a shows the continuous wave EPR

spectrum, in the dark and at room temperature, on a representative ternary blend sample of weight ratio 0.10:0.90 of ZnPc:$F_8$ZnPc doped with 0.05 MR of $F_6$-TCNNQ. The EPR spectrum shows a signal with narrow line width and a field position typical of organic radicals[22]. From the spectral simulations, shown as dashed lines in Fig. 2a, two species are found to contribute to the signal with $g$-values of $g_A = 2.0023 \pm 0.0005$ and $g_B = 2.0033 \pm 0.0005$. These $g$-values are similar to those previously reported for positive polarons localised on ZnPc ($g_A$) and the radical anion of $F_4$-TCNQ, which we assume is comparable to the $g$-value of $F_6$-TCNNQ ($g_B$)[23–25]. This suggests that, in agreement with the PDS, there is a ground-state electron transfer from the host blend to the dopant.

Similar EPR analysis is performed for hosts with a range of blend ratios of ZnPc to $F_8$ZnPc, all doped with 0.05 MR $F_6$-TCNNQ. The spectra are shown in Fig. 2b. There is a clear increase in the absolute EPR intensity as the content of ZnPc increases. The normalised spectra and simulations for each blend are reported in Supplementary Fig. 3 with the $g$-values and line widths summarised in Supplementary Table 2. The fits reveal the presence of the same two species as detected in Fig. 2a, for all blends except that of the pure $F_8$ZnPc as the host. For the case of $F_8$ZnPc, shown in Fig. 2c, the EPR signal is less intense, which indicates that the sample contains fewer unpaired spins. Although this signal is negligible compared to the ZnPc signal, it confirms that there is charge transfer between the $F_8$ZnPc and the dopant, as observed in the PDS, despite the nominally unfavourable energetic offset. The ground-state charge transfer from host blend to dopant is dominated by the interaction between the ZnPc and dopant.

The EPR intensity is proportional to the overall amount of unpaired spins—a sum of contributions from both ground-state integer-charge transfer complexes (ICTCs) and mobile charges on the ZnPc and $F_8$ZnPc in the sample. The absolute number of

spins in the samples can be estimated by double integration of the EPR signal (absolute EPR intensity) compared to a reference sample with a known number of spins. In the mixed host cases, the EPR line has an even contribution of holes on the ZnPc ($p$) and $F_6$-TCNNQ anions ($N_A^-$)[26], neglecting the small contribution of polarons on $F_8$ZnPc ($q$). Therefore, the polaron concentration is the total amount of measured spins divided by twice the sample volume. Finally, the doping efficiency is determined as the measured polaron concentration over the total number of dopants in the film ($p + q/N_A$), with the number of dopants measured during deposition using a quartz crystal microbalance.

In Fig. 2d, the doping efficiency, as determined from the EPR signal, is plotted against ratio of ZnPc in the host blend. There is a clear monotonic increase in the doping efficiency as the content of ZnPc increases, with the highest doping efficiency of (31.9 ± 5.7)%, reached for the host of pure ZnPc. This less than unity doping efficiency suggests that at 0.05 MR, we are in the dopant reserve regime, in agreement with UPS measurements on the same system[11]. The contribution from the $F_8$ZnPc polarons (EPR species C) gives the lowest doping efficiency of (1.3 ± 0.2)%. The overall trend of the doping efficiency matches that observed in the PDS for the dopant anion peak, as expected due to charge conservation.

**Statistical model**. To establish how the trend in doping efficiency is related to the energy levels of the ternary blend system, we apply a statistical description based on classical semiconductor theory. This approach has been used to attribute shifts in the Fermi level of doped organic semiconductors to three regimes—trap filling (at doping concentrations below MR < $10^{-4}$), dopant saturation and dopant reserve (MR > $10^{-3}$)[11,27]. More recently the efficient dissociation of ICTCs was explained by adjusting the energetic disorder in the model[8]. For our case at 0.05 MR, we expect to operate in the dopant reserve regime, where trap states

intrinsic to organic semiconductors have a negligible impact on doping efficiency[11].

Our statistical model contains an additional energy level for the third component in the ternary blend. The levels are approximated as Gaussian, all with the same standard deviation of $\sigma$. The host blend comprises two separate density of states (DOS): one at higher energy representing the ZnPc and one at lower energy for the $F_8$ZnPc. The centre of the dopant's DOS ($E_A$) is defined with respect to the centre of the ZnPc DOS. The occupation of each of the levels is determined by Fermi-Dirac integrals, with the Fermi level set by numerically solving the neutrality condition

$$p + q = N_A^-, \tag{1}$$

where $p$ and $q$ are the number of holes (both mobile and bound) residing on ZnPc and $F_8$ZnPc, respectively, and $N_A^-$ is the number of ionised dopants. Further details of the calculations can be found in Supplementary Note 2.

To model the ternary blend, we consider the effects of the quadrupole interactions between the host semiconductors, ZnPc and $F_8$ZnPc. The consequence of this interaction is twofold. Firstly, the difference in IP between the two components becomes considerably smaller, as compared to their IP in pristine films. Secondly, both IPs shift linearly dependent on blend ratio. The energy levels of the two hosts in the mixed films are taken from the UPS measurements published by Schwarze et al.[1], with the gradient of the shift determined by linear fits, as shown in Supplementary Fig. 4. Whether the acceptor level of the dopant is affected by the same mechanism is an open question. We, therefore, investigate both possibilities for the DOS of the system —one with a fixed dopant level (Fig. 3a), and one with the dopant energy level shifting (Fig. 3b).

In Fig. 3a, b, the area under each of the host components is proportional to the content in the host blend. The area of the dopant DOS is constant, as the doping concentration is fixed. The Fermi level, calculated via Eq. (1), is marked by a dashed orange

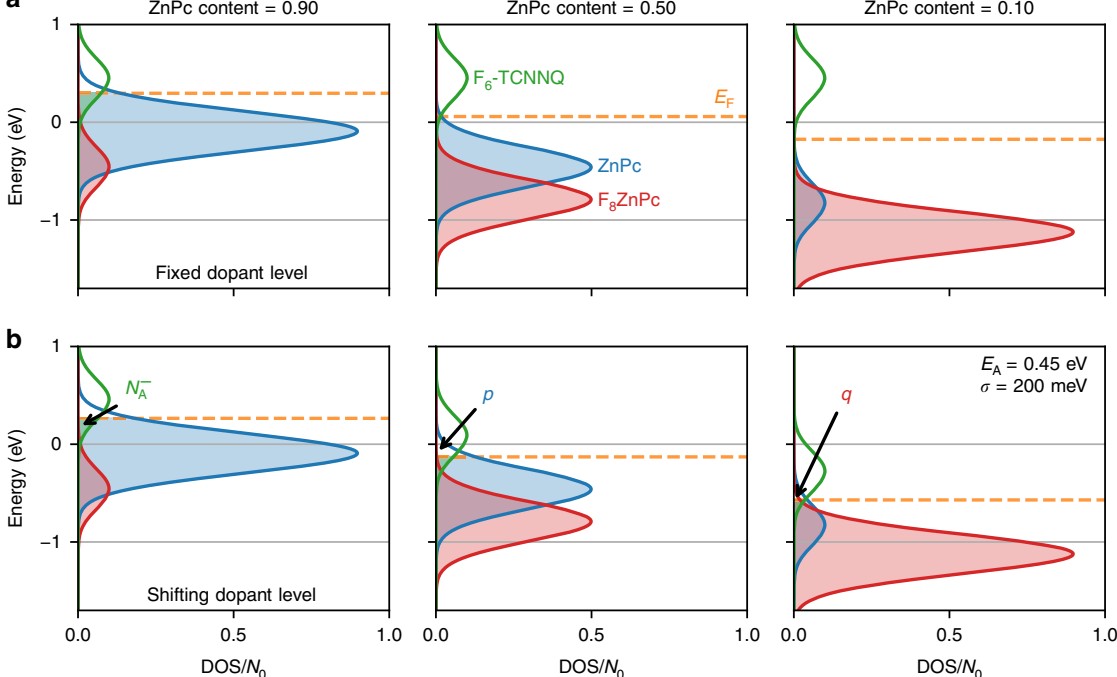

**Fig. 3** Density of states for the statistical model. The density of states (DOS) for the two cases of distributions: **a** with the dopant level at fixed energy and **b** with a dopant level shifting in energy, for molar ratios 9:1, 1:1, 1:9 of ZnPc:$F_8$ZnPc (left to right). The Fermi level $E_F$ is calculated via Eq. (1) and is plotted as a dashed orange line. Below the Fermi level, the dopants are ionised (filled green colour) and the host states are full (filled blue and red). The model parameters are reported inset in the bottom right panel, and are varied in the analysis later.

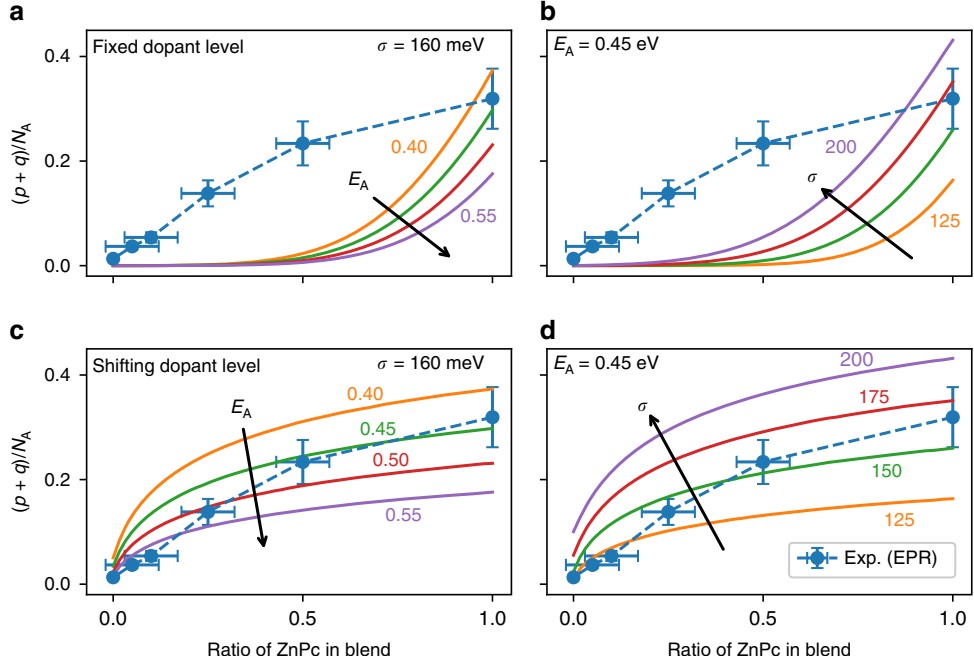

**Fig. 4** Statistical model comparison to EPR. Comparisons of the experimentally measured doping efficiency and the statistical models against weight ratio of ZnPc in the host blend, with the centre of the dopant's DOS $E_A$ and standard deviation $\sigma$ varied independently. For the case where the dopant level is fixed: **a** shows variations in $E_A$ with $\sigma = 160$ meV and **b** shows variations in $\sigma$ with $E_A = 0.45$ eV. For the case where the dopant energy level shifts: **c** shows the variation in $E_A$ with $\sigma = 160$ meV and **d** variations in $\sigma$ with $E_A = 0.45$ eV. The doping efficiency as determined via EPR is shown on all figures, with the error bars representing the uncertainty in the spectral simulations and material deposition rates.

line. All states below the Fermi level are occupied (shaded) whilst the states above are unoccupied (unshaded). Figure 3a shows the first case with the position of dopant energy level fixed. The Fermi level shifts to lower energy as the content of ZnPc in the host blend decreases (left to right). Consequently, the proportion of ionised dopants $N_A^-$ decreases with ZnPc content. Figure 3b shows the second DOS model, where we consider the impact of electrostatic interactions arising from the quadrupole moments on the dopant's energy levels. The major consequence is that $E_A$ shifts downwards in energy, with respect to the vacuum level, dependent on the blend ratio of ZnPc and $F_8$ZnPc.

GIWAXS images of the doped films are presented in Supplementary Fig. 2, and indicate a preference for edge-on orientation. Previous investigations have shown that molecular orientation can play a significant role in the electrostatic interactions[28,29], and determines which component of the quadrupole tensor is dominant[30,31]. For planar molecules, such as ZnPc, with edge-on orientation, the largest component of the quadrupole interaction is along the $\pi$-$\pi$ stacking direction ($Q_\pi$), corresponding to the shortest intermolecular distance[16]. As the addition of the p-dopant at low concentrations does not significantly impact the host packing[32,33], it is expected that $Q_\pi$ remains the dominant component in the doped films. Density functional theory (DFT) calculations show that the dopant carries a net positive $Q_\pi$, comparable to that of the $F_8$ZnPc (see Supplementary Table 3 and Supplementary Fig. 5). Therefore, a coherent superposition of the dopant's quadrupolar molecular field with the ZnPc will result in a similar effect to that observed for the $F_8$ZnPc—a linear shift in the dopant's energy dependent on the content of ZnPc in the host blend. As the doping concentration is low, we expect little impact of the dopant's quadrupole moment on either of the host molecules. To a first approximation, the gradient of the dopant's shift in energy is set equal to that of the host distributions.

We compare the two statistical descriptions to the experimentally determined doping efficiency, plotted in Fig. 4. The results of parameter testing $E_A$ and $\sigma$ are shown as solid lines in Fig. 4 for both cases of a fixed and shifting dopant level. For the fixed dopant case in Fig. 4a, b, the doping efficiency only begins to increase from zero when more then approximately 50% of the host blend is ZnPc. Above 50%, the doping efficiency rises quickly with a strong dependence on $E_A$. Overall, the trends seen in the experimental data are difficult to reproduce with physical parameters for $E_A$ and $\sigma$.

In contrast, the doping efficiency for the case with a shifting dopant level in Fig. 4c, d shows a trend in close agreement with the experimental data. Upon introduction of ZnPc into the host blend, the doping efficiency steeply rises. At high ZnPc content ratios, the efficiency tends toward a saturation value. With larger $E_A$, the doping efficiency is significantly reduced. In both models, the doping efficiency improves with increasing $\sigma$ (Fig. 4b, d). As noted in recent studies, this highlights the importance of considering not just the energetic offset between host and dopant, but also the energetic disorder of the system as this can severely impact the doping efficiency[8,9,34].

The statistical model can be extended to separate the number of charge carriers $p$ in to the contributions of mobile carriers $p_{mob}$ and those bound in ICTCs $N_{CT}^+$ (see Supplementary Note 3). Introducing ICTCs with binding energy $E_{CT}^b$ has no impact on dopant ionisation $N_A^-$, as can be seen in Supplementary Fig. 6. Instead with larger $E_{CT}^b$, the primary effect is that a greater proportion of charge carriers remain bound in ICTCs.

Finally, we comment on the placement of dopant's DOS. Experimental measurements on unmixed thin films, place the EA of $F_6$-TCNNQ (5.5 eV) below that of the IP of ZnPc (5.1 eV)[1,12]. However, to simulate physically realistic doping efficiencies below 100%, the statistical descriptions require that the dopant's DOS is above that of the host's DOS. As the doping mechanism between

ZnPc:$F_6$-TCNNQ is reported as integer-charge transfer[8,12], we cannot explain the position of $E_A$ with reference to hybridised antibonding states[11,35]. Instead, we propose that the dopant's energy level shifts above that of the hosts as a consequence of the electrostatic interactions in the mixed films, in accordance with recent calculations[36]. For $F_8$ZnPc in a mixed film with ZnPc, the IP shifts by 0.86 eV relative to a $F_8$ZnPc pristine film[1]. A similar shift places the dopant's acceptor level from 0.40 eV below ZnPc (the difference between their pristine IP and EA) to 0.46 eV above the centre of the ZnPc DOS. With the disorder described by a standard deviation $\sigma$ around 160–180 meV, as measured for ZnPc:$F_6$-TCNNQ by UPS measurements[8], this value for the dopant's acceptor level is in close agreement with our statistical description. As further validation, we perform EPR measurements at low temperature $T = 80$ K. In Supplementary Fig. 7, we are able to reproduce the low temperature experimental results using our model with $E_A$ and $\sigma$ kept constant. The doping efficiency decreases at lower temperature because fewer holes possess the required thermal energy to bridge the gap between the host and dopant energy levels. Overall, considering quadrupole interactions between host and dopant yields a consistent picture of the doping process in the ZnPc:$F_6$-TCNNQ system.

## Discussion

We establish that the electrostatic interactions, which allow for the continuous tuning of effective energy levels, profoundly effect the doping process in organic semiconductors. The role of energy levels in the doping process is investigated using a ternary blend system comprising hosts ZnPc and $F_8$ZnPc and dopant $F_6$-TCNNQ. Through PDS and EPR measurements, a monotonic increase in the doping efficiency is found from $(1.3 \pm 0.2)$% to $(31.9 \pm 5.7)$% as the content of ZnPc in the blend ratio of ZnPc:$F_8$ZnPc increases. We numerically solve the neutrality equation for the ternary system and find that the trend of the experimental data is reproduced only by including a shift in the dopant's acceptor level. We explain this shift by considering the effect of electrostatic interactions between the hosts and dopant molecules.

The simplified picture of selecting hosts and dopants with favourable energy level offset, as dictated by measurements or calculations of their IP and EA in pristine films, must be extended to include electrostatic effects arising in mixed films, and explains why some host dopant combinations with unfavourable offset still show the doping effect. Future dopant design should consider the quadrupole moments of both the dopant molecule itself, and the host in which it is to be employed, as important molecular parameters, such that doping efficiency and device performance can be maximised. Finally, we propose that ternary blends of organic semiconductors with energy levels subject to the electrostatic shifts observed here, allow for simultaneous control of the molecular energy levels and Fermi level, paving the way for improved performance of organic optoelectronic devices.

## Methods

**Sample fabrication**. All samples are prepared by vacuum deposition with the dopant $F_6$-TCNNQ (1, 3, 4, 5, 7, 8-hexafluoro-tetracyanonaphthoquinodimethane, purchased from Novaled GmbH) concentration fixed at 0.05 molar ratio (MR) whilst the fraction between the two host molecules ZnPc (zinc-phthalocyanine) and its octuply fluorinated derivative, $F_8$ZnPc (both purchased in sublimed grade from Lumtec Corp.), is varied. The total deposition rate of the host semiconductor(s) is kept constant at $(0.40 \pm 0.02)$ Å/s. It is assumed that the ZnPc and $F_8$ZnPc have a similar density of 1.55 g cm$^{-3}$. Furthermore, it is assumed that the dopant replaces host molecules in the films preserving the overall density such that $\rho_{dopant} = \rho_{host}$. Prior to deposition, all the substrates are cleaned for 10 min in an ultrasonic bath of 2.5% Hellmanex solution, followed by DI water, acetone and finally isopropyl alcohol. The substrates are treated with $O_2$ plasma for 10 min before being loaded into the vacuum chamber.

**GIWAXS**. Grazing-incidence wide-angle x-ray scattering (GIWAXS) studies are carried out at the Surface and Interface Diffraction beamline (I07) at the Diamond Light Source (DLS) using a beam energy of 20 keV (0.62 Å) and a Pilatus2M area detector. The samples are probed while inside a vacuum deposition chamber at a pressure of around $10^{-3}$ mbar with the MINERVA setup[37]. The sample-to-detector distance was 42.1 cm as determined via AgBeh calibration. Images are converted to 2D reciprocal space using the DAWN software package with an applied polarisation and solid angle correction[38].

**Photothermal deflection spectroscopy**. Films of 40 nm thickness are deposited on quartz substrates and mounted in a cuvette containing the liquid $CCl_4$. The PDS data is compared and scaled using transmission-reflection measurements, in the energy range of strong absorption (the first Q-band peak of zinc-phthalocyanine).

**Electron paramagnetic resonance spectroscopy**. Electron paramagnetic resonance (EPR) measurements are made on thin films of 50 nm thickness deposited on microscope cover glass, cut to a width of 3 mm with a diamond tipped glass cutter. The samples are placed in quartz EPR tubes which are sealed in a nitrogen glovebox, such that all EPR measurements are made without air exposure. The continuous wave EPR spectra are recorded on a Bruker Elexsys E680 X-band spectrometer with a nitrogen gas-flow cryostat for sample temperature control. The experimental parameters were set with an amplitude modulation = 1 G and microwave power = 0.2 mW (30 dB attenuation). Spectral simulations are performed using routines of Easyspin[39].

## Data availability

All data is available on our research group's website [https://github.com/AFMD] or from the corresponding authors upon request.

## Code availability

The statistical model code is available on our research group's website [https://github.com/AFMD] or from the corresponding authors upon request.

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

## Acknowledgements

R.W. was supported by the UK Engineering and Physical Sciences Research Council (EPSRC) grant EP/L016702/1 for the Centre for Doctoral Training in Plastic Electronics. A.P. acknowledges the European Union's Horizon 2020 research and innovation programme under Marie Sklodowska Curie Grant agreement No. 722651 (SEPOMO project). P.K. thanks the Global Challenges Research Fund from the UK Science and Technology Facilities Council (STFC), grant number ST/R002754/1: Synchrotron Techniques for African Research and Technology (START). A.E.L. thanks the EPSRC for funding through the Doctoral Training Partnership EP/N509711/1 as well as the STFC, ISIS Neutron and Muon facility and project 1948713. Further support was through funding from the European Union in FP7 (Project ID 630864) and the STFC Challenge-Led Applied Systems Programme (CLASP, ST/L003309/1) focused on advancing the commercialisation of organic solar cells. EPR measurements were performed in the Centre for Advanced ESR (CAESR), located in the Department of Chemistry of the University of Oxford, and this work was supported by the EPSRC (EP/L011972/1). A.P. thanks Dr William Myers, CAESR facility, for his kind assistance with the EPR measurements. The GIWAXS data was collected during experiment SI20426-1 at beamline I07 of the Diamond Light Source. We are grateful to J. Naylor, D. Wicks and A. Dorman of K.J. Lesker Ltd. for providing deposition control and evaporation sources along with technical support for MINERVA. Finally, the authors thank Dr Ivan Ramirez for useful discussions and guidance on sample preparation.

## Author contributions

The original draft of the paper was written by R.W. with all authors contributing to reviewing and editing. R.W. prepared the samples, analysed the PDS, wrote the code for the statistical simulations, took the GIWAXS images, and made the DFT calculations. A.P. performed the EPR experiments and analysis. A.E.L. supervised the diffraction experiments and analysed the GIWAXS images. O.T. made the PDS measurements. The project was conceived by R.W., M.R. and J.N., with significant input along the way from A.P. and P.K. J.N. and M.R. provided project supervision.

## Competing interests

The authors declare no competing interests.
