## [Peer Review File · Nature Communications]

Reviewers' comments:

Reviewer #1 (Remarks to the Author):

Warren et al. describe in their manuscript "Controlling molecular energy levels and Fermi level - towards fully tailored energetics in organic semiconductors" a detailed study of doping in a mixed system of the organic semiconductors ZnPc and F8ZnPc. The work is based on a result of the Leo group showing that the level energy of this mixed system can be tuned by adapting the mixing ration. Therefore, by electrically doping this system, the authors are able to not only tune the Fermi energy inside their system (by controlling the concentration of the electric dopant), but the energy levels as well.

Scientifically, the manuscript shows that the dopant level is shifted in the same way as the energy levels of the host mixture, i.e. the dopant activation energy shifts with the mixing ratio of ZnPc and F8ZnPc. This result is interpreted in terms of a quadrupole electrostatic interaction between dopants and host materials that is as well thought to cause the shift in the energy levels of the host in first place. The authors claim that this long range electrostatic interaction is crucially determining the energy of the dopant in the film.

The manuscript is very well written and the experiments are of a very high quality. The novelty of the manuscript is good.

In addition, I have the following comments:

- the authors should discuss in more detail the contribution N_{CT} of the charge neutrality equation (1). How has the energy of the ICTC (I assume that integer charge transfer complexes) been fixed (g_{CT} and E^b_{CT})? Does this energy shift with the mixing ratio of the host materials similar to the doping energy? Overall, treating this parameter as a fitting parameter might be crucial.
- Please provide a table of all parameters used to model doping as e.g. in Fig. 3 and 4
- The argument of the authors relies on the fitting results assuming a constant or a shifting dopant level. Providing a direct measurement of the dopant activation energy and the observed shift would greatly strengthen their argument. Possible measurement techniques are temperature dependent impedance spectroscopy/CV measurement. Temperature dependent conductivity measurements should provide another hint.

Smaller comments:

- page 2, third paragraph: two "the" in the beginning
- The two fits in Fig. 4 fit the doping efficiency using different ranges of the activation energy, which makes a direct comparison difficult. Can the authors extend the fitting range?

Bjorn Lussem

Reviewer #2 (Remarks to the Author):

The Fermi level of organic semiconductor materials plays a decisive role for a variety of electronic properties. Doping is one way to tune this energy relative to the semiconductor's HOMO or LUMO levels. In their manuscript 'Controlling molecular energy levels and Fermi level - towards fully tailored energetics in organic semiconductors', Warren and co-workers study the influence of the composition of the host material ZnPc/F8ZnPc on the doping efficiency of the p-type dopant F6TCNNQ.

The presented findings are quite consistent with recent literature. Moreover, conclusions such as "This tells us that the simple picture of selecting hosts and dopants with favourable energy-level offset, as dictated by measurements or calculations of their IP and EA in pristine films, must be extended to account for electrostatic effects in the mixed films, and explains why some host-dopant combinations with unfavourable offset still show the doping effect" are not very surprising, given the latest works in literature, but rather expected.

These sections do not convince me of sufficient level of novelty for Nature Communications. In addition, there are some concerns which I list here below.

There is also a question regarding the charge neutrality equation. I do not understand the definition of NCT+. There are three species but it is not clear where these positive charges are located. The text says „which remain bound in ICTCs“ but since p and q are already defined as holes residing on ZnPc and F8ZnPc, the remaining option is that the positive charges NCT+ are located at the dopants. This however, is very unlikely because F6TCNNQ is a p-dopant. The authors need to clarify this point.

An issue that might be related appears in Fig. 3. For instance in Fig. 3a, right panel, there seems to be excess density of electrons at the dopant which is indicated by the position of the Fermi level relative to the green DOS. The position of the Fermi level also indicates that the DOS of both phthalocyanines are basically completely filled. This implies a net charge and not charge neutrality. There is either a mistake or the models need better explanation or referencing. This inconsistency might affect the validity of the conclusions drawn from the study in Fig.3 as well.

Finally, the statement on the donor's acceptor level $EA=0.46$ eV is unclear. Is it $EA=-0.46$ eV or did the authors change the reference energy?

Reviewer #3 (Remarks to the Author):

This manuscript deals with a very important question for the field of organic semiconductors: the control of the energy levels and their doping. These are fundamental for a tailored design in view of particular applications. Moreover, to the best of my knowledge Warren et al. are the first to combine to approaches that have been put forward recently, and in this way they substantially advance the field.

The presented experimental data are sound, the systematic studies are of high quality, the applied model is convincing! It is demonstrated that level control and doping can be achieved in a desired manner, and - importantly - it is explained on a model basis how the mechanism might work. This will help others to further develop devices and theoretical descriptions.

I fully welcome this manuscript and the presented results and their discussion - and to my opinion the manuscript deserves publication in Nature Communications.

I have one point which I would like the authors to consider:

What is the role of the mutual orientation of the host and dopant molecules? How would one try to take this into account? This mutual orientation certainly changes the electrostatic interactions which are a main focus of the considerations and it would be instructive to learn about them more.

We thank the three referees for their comments which helped us to improve the manuscript. In the following, we address all the comments (kept in *italics*) point-by-point and highlight the changes made in the manuscript.

Reviewer #1 (Remarks to the Author):

Warren et al. describe in their manuscript "Controlling molecular energy levels and Fermi level - towards fully tailored energetics in organic semiconductors" a detailed study of doping in a mixed system of the organic semiconductors ZnPc and F8ZnPc. The work is based on a result of the Leo group showing that the level energy of this mixed system can be tuned by adapting the mixing ration. Therefore, by electrically doping this system, the authors are able to not only tune the Fermi energy inside their system (by controlling the concentration of the electric dopant), but the energy levels as well.

Scientifically, the manuscript shows that the dopant level is shifted in the same way as the energy levels of the host mixture, i.e. the dopant activation energy shifts with the mixing ratio of ZnPc and F8ZnPc. This result is interpreted in terms of a quadrupole electrostatic interaction between dopants and host materials that is as well thought to cause the shift in the energy levels of the host in first place. The authors claim that this long range electrostatic interaction is crucially determining the energy of the dopant in the film.

The manuscript is very well written and the experiments are of a very high quality. The novelty of the manuscript is good.

In addition, I have the following comments:

(1) *The authors should discuss in more detail the contribution N_{CT} of the charge neutrality equation (1). How has the energy of the ICTC (I assume that integer charge transfer complexes) been fixed (g_{cT} and E^b_{CT})? Does this energy shift with the mixing ratio of the host materials similar to the doping energy? Overall, treating this parameter as a fitting parameter might be crucial.*

Response: We thank the reviewer for raising a very interesting point. Understanding the role of integer charge transfer complexes (ICTC) formation is important to determine the number of mobile carriers produced in the doping process.

In combination with Reviewer #2's comments, we have improved the discussion of the statistical model and made the explanations (Supplementary Notes 2 and 3) clearer. The ICTC binding energy E^b_{CT} plays an important role in determining the proportion of mobile charge carriers to those that remain bound in ICTCs.

In Supplementary Figure 6, we plot the contributions of bound ICTCs (N_{CT}^+) and mobile charges (p_{mob}) that sum to the total number of charge (p) residing on the host semiconductor (the contribution of charges on F₈ZnPc has been found to be very small and has therefore been omitted). With increasing E^b_{CT} , the number of ionised dopants (N_A^-) remains unaffected. However from Supplementary Figure 6, it is clear that the proportion of mobile carriers to bound carriers decreases.

As EPR measures the sum of the contributions from both mobile and bound carriers, variations in E^b_{CT} do not impact our conclusions. However the model is now more complete, with the parameter study of E^b_{CT} added to the Supplementary Information (Supplementary Figure 6).

Manuscript:

- Updated the explanation of the model in Supplementary notes 2 and 3.
- Added parameter study of E_{CT}^b shown in Supplementary Figure 6.
- Added to main text: “The statistical model can be extended to separate the number of charge carriers p in to the contributions of mobile carriers p_{mob} and those bound in ICTCs N_{CT}^+ (see Supplementary Note 3). Introducing ICTCs with binding energy E_{CT}^b has no impact on dopant ionisation N_A^- , as can be seen in Supplementary Figure 6. Instead with larger E_{CT}^b , the primary effect is that a greater proportion of charge carriers remain bound in ICTCs.”

(2) Please provide a table of all parameters used to model doping as e.g. in Fig. 3 and 4

Response: We agree and have re-designed Figs. 3 and 4 to include the simulation parameters within the figure panels. Further in each panel of Fig. 4, we have added arrows to highlight which fit parameter is being varied.

Additionally, we would like to highlight that the full simulation code will be available on GitHub (<https://github.com/AFMD>) in due course. This contains all the parameters and data generated from the statistical model.

Manuscript:

- Updated Figure 3 to show model parameters kept constant (right hand panel Fig. 3b)
- Updated Figure 4 to show model parameters kept constant (inset Fig. 4a-d)

(3) The argument of the authors relies on the fitting results assuming a constant or a shifting dopant level. Providing a direct measurement of the dopant activation energy and the observed shift would greatly strengthen their argument. Possible measurement techniques are temperature dependent impedance spectroscopy/CV measurement. Temperature dependent conductivity measurements should provide another hint.

Response: We agree with the reviewer that a temperature dependent measurement would strengthen our argument. Impedance spectroscopy/CV are one option, but we can also test the activation energy using temperature dependent EPR. Using EPR has the benefit that no metal contacts are needed, which would complicate the interpretation. Therefore, we perform further EPR experiments at $T = 80$ K. To test our dopant activation level, we keep it constant in the model and vary only the temperature, setting it to 80 K. The results are presented in Supplementary Figure 7 and show that our model’s dopant activation energy fits the low temperature experimental data well.

Manuscript:

- Added EPR measurements at low temperature (Supplementary Figure 7).
- Added: “ To further validate our value of the dopant acceptor level, we perform EPR measurements at low temperature $T = 80$ K. In Supplementary Figure 7, we are able to reproduce the low temperature experimental results using our model with E_A and σ kept constant. The doping efficiency decreases at lower temperature because fewer holes possess the required thermal energy to bridge the gap between the host and dopant energy levels.”

Smaller comments:

(4) page 2, third paragraph: two "the" in the beginning

Manuscript:

- Changed: “Tuning the ~~the~~ energetic offset between host [...]”

(5) *The two fits in Fig. 4 fit the doping efficiency using different ranges of the activation energy, which makes a direct comparison difficult. Can the authors extend the fitting range?*

Manuscript:

- Updated Figure 4 to show the same fitting ranges across the two models.

Reviewer #2 (Remarks to the Author):

The Fermi level of organic semiconductor materials plays a decisive role for a variety of electronic properties. Doping is one way to tune this energy relative to the semiconductor’s HOMO or LUMO levels. In their manuscript ‘Controlling molecular energy levels and Fermi level - towards fully tailored energetics in organic semiconductors’, Warren and co-workers study the influence of the composition of the host material ZnPc/F8ZnPc on the doping efficiency of the p-type dopant F6TCNNQ.

The presented findings are quite consistent with recent literature.

(1) *Moreover, conclusions such as “This tells us that the simple picture of selecting hosts and dopants with favourable energy-level offset, as dictated by measurements or calculations of their IP and EA in pristine films, must be extended to account for electrostatic effects in the mixed films, and explains why some host-dopant combinations with unfavourable offset still show the doping effect” are not very surprising, given the latest works in literature, but rather expected.*

These sections do not convince me of sufficient level of novelty for Nature Communications. In addition, there are some concerns which I list here below.

Response: With renewed interest in both molecular doping (DOI: 10.1038/s41467-018-03302-z) and the impact of quadrupole interactions (DOI: 10.1126/science.aaf0590), our manuscript is the first to combine these effects. Summarising the originality: we present a novel ternary system, a new method for probing the ionised dopants (EPR), and, finally, develop a consistent model to explain our results. More realistic guidelines for selecting suitable combinations of host (blends) and dopant molecules directly follow from our findings. The results also highlight paths for technological progress for functional devices such as solar cells and LEDs. For example, the presented superior control of an organic semiconductor’s energy levels, allows to accurately tailor contact properties with a higher degree of freedom than before. However, we do agree that the sentence could be improved to better highlight the novelty of the manuscript.

Manuscript:

- The sentence has been rephrased to read: “The simplified picture of selecting hosts and dopants with favourable energy-level offset, as dictated by measurements or calculations of their IP and EA in pristine films, must be extended to include electrostatic effects arising in mixed films, and explains why some host-dopant combinations with unfavourable offset still show the doping effect.”

(2) *There is also a question regarding the charge neutrality equation. I do not understand the definition of NCT+. There are three species but it is not clear where these positive charges are located. The text says „which remain bound in ICTCs“ but since p and q are already defined as holes residing on ZnPc and F8ZnPc, the remaining option is that the positive charges NCT+ are*

located at the dopants. This however, is very unlikely because F6TCNNQ is a p-dopant. The authors need to clarify this point.

Response: We agree that more clarity is needed, and in combination with Reviewer #1's comments, we have extended the explanation of the model and separated the total charge carriers p on the hosts in to two contributions from mobile charges p_{mob} and those that are bound in ICTCs N_{CT}^+ .

Manuscript:

- Updated Equation 1: $p + q = N_A^-$, and split total charge carriers in to two contributions $p = p_{\text{mob}} + N_{\text{CT}}^+$ (assuming a negligible contribution from q)
- Supplementary notes 2 and 3 have been updated to clarify the model.

(3) An issue that might be related appears in Fig. 3. For instance in Fig. 3a, right panel, there seems to be excess density of electrons at the dopant which is indicated by the position of the Fermi level relative to the green DOS. The position of the Fermi level also indicates that the DOS of both phthalocyanines are basically completely filled. This implies a net charge and not charge neutrality. There is either a mistake or the models need better explanation or referencing. This inconsistency might affect the validity of the conclusions drawn from the study in Fig.3 as well.

Response: Many thanks for picking this up. We have replaced Figure 3 with the correct version. The simulations in Figure 4 were not affected.

We would like to highlight that all the code will be available on GitHub (<https://github.com/AFMD>) in due course.

Manuscript:

- Right panel of Figure 3b updated.

(4) Finally, the statement on the donor's acceptor level $EA=0.46$ eV is unclear. Is it $EA=-0.46$ eV or did the authors change the reference energy?

Response: We agree with the reviewer that position of the reference energy was confusing. We have updated the sentence to explicitly say that the dopant's acceptor level is 0.46 eV above the centre of the ZnPc DOS.

Manuscript:

- Main text updated to read: "For $F_8\text{ZnPc}$ in a mixed film with ZnPc, the IP shifts by 0.86 eV relative to a $F_8\text{ZnPc}$ pristine film. A similar shift places the dopant's acceptor level from 0.40 eV below ZnPc (the difference between their pristine IP and EA) to 0.46 eV above the centre of the ZnPc DOS."

Reviewer #3 (Remarks to the Author):

This manuscript deals with a very important question for the field of organic semiconductors: the control of the energy levels and their doping. These are fundamental for a tailored design in view of particular applications. Moreover, to the best of my knowledge Warren et al. are the first to combine to approaches that have been put forward recently, and in this way they substantially advance the field.

The presented experimental data are sound, the systematic studies are of high quality, the applied model is convincing! It is demonstrated that level control and doping can be achieved in a desired

manner, and - importantly - it is explained on a model basis how the mechanism might work. This will help others to further develop devices and theoretical descriptions.

I fully welcome this manuscript and the presented results and their discussion - and to my opinion the manuscript deserves publication in Nature Communications.

I have one point which I would like the authors to consider:

(1) What is the role of the mutual orientation of the host and dopant molecules? How would one try to take this into account? This mutual orientation certainly changes the electrostatic interactions which are a main focus of the considerations and it would be instructive to learn about them more.

Response: We thank the reviewer for the positive comments and agree that the mutual orientation of the host and dopant molecules will play a role in the electrostatic interactions. Indeed, for ZnPc, Schwarze et al. (DOI: 10.1038/s41467-019-10435-2) recently reported that the IP in edge-on orientation was found to be 0.22 eV smaller than face-on orientation. Similarly F₈ZnPc was found to have an IP dependent on orientation, but with an opposite shift of 0.37 eV from face-on to edge-on orientation. The shifts in IP were found to correlate well with the magnitude and sign of the quadrupole moment in the π - π stacking direction (Q_{π}), with the trend extending to a dozen other systems of planar molecules.

For our case with a low p-doping concentration, we expect that the addition of the dopant does not interrupt the host packing. The GIWAXS images in Supplementary Figure 2 show similar features to the films without the dopant (DOI: 10.1063/1.5080505).

Furthermore, F4-TCNQ was shown to adopt a co-facial mutual orientation with the hosts (DOI: 10.1038/ncomms9560). As F6-TCNNQ has a very similar molecular structure to F4-TCNQ, we therefore assume a similar arrangement. Overall, this is in agreement that the Q_{π} component of the dopant quadrupole moment is dominant.

We have highlighted in the manuscript the dependence on orientation and added suitable references.

Manuscript:

- Added: “Previous investigations have shown that molecular orientation can play a significant role in the electrostatic interactions, and determines which component of the quadrupole tensor is dominant.”
- “For planar molecules, such as ZnPc, with edge-on orientation, the largest component of the quadrupole interaction is along the π - π stacking direction (Q_{π}), corresponding to the shortest intermolecular distance.”
- “As the addition of the p-dopant at low concentrations does not significantly impact the host packing, it is expected that Q_{π} remains the dominant component in the doped films.”

Best regards,

Ross Warren and Moritz Riede

REVIEWERS' COMMENTS:

Reviewer #1 (Remarks to the Author):

The authors have successfully addressed all my comments.

Reviewer #2 (Remarks to the Author):

In the revised manuscript the authors have clarified the open questions, have corrected the figure and have revised some critical sentences. I think it can be published.

Reviewer #3 (Remarks to the Author):

I carefully went through the authors' response to all points raised by the Reviewers in the previous round. To my opinion, the response is complete and the manuscript has been improved in course of the points raised.

I now recommend publication of this work.